# Advanced Farming Strategies Using NASA POWER Data in Peanut-Producing Regions without Surface Meteorological Stations

Thiago Orlando Costa Barboza [1,*], Marcelo Araújo Junqueira Ferraz [1], Cristiane Pilon [2], George Vellidis [2], Taynara Tuany Borges Valeriano [3] and Adão Felipe dos Santos [1,*]

1. Department of Agriculture, School of Agriculture Sciences of Lavras, Federal University of Lavras (UFLA), Lavras 37200-900, MG, Brazil; marcelo.ferraz1@estudante.ufla.br
2. Department of Crop and Soil Sciences, University of Georgia, Tifton, GA 31793, USA; cpilon@uga.edu (C.P.); yiorgos@uga.edu (G.V.)
3. Bayer CropScience, 51368 Leverkusen, Germany; taynarabvaleriano@gmail.com
* Correspondence: thiagocostaagro@gmail.com (T.O.C.B.); adao.felipe@ufla.br (A.F.d.S.)

**Abstract:** Understanding the impact of climate on peanut growth is crucial, given the importance of temperature in peanut to accumulate Growing Degree Days (GDD). Therefore, our study aimed to compare data sourced from the NASA POWER platform with information from surface weather stations to identify underlying climate variables associated with peanut maturity (PMI). Second, we sought to devise alternative methods for calculating GDD in peanut fields without nearby weather stations. We utilized four peanut production fields in the state of Georgia, USA, using the cultivar Georgia-06G. Weather data from surface stations located near peanut fields were obtained from the University of Georgia's weather stations. Corresponding data from the NASA POWER platform were downloaded by inputting the geographic coordinates of the weather stations. The climate variables included maximum and minimum temperatures, average temperature, solar radiation, surface pressure, relative humidity, and wind speed. We evaluated the platforms using Pearson correlation (r) analysis ($p < 0.05$), linear regression analysis, assessing coefficient of determination ($R^2$), root mean square error (RMSE), and Willmott index (d), as well as principal component analysis. Among the climate variables, maximum and minimum temperatures, average temperature, and solar radiation showed the highest $R^2$ values, along with low RMSE values. Conversely, wind speed and relative humidity exhibited lower correlation values with errors higher than those of the other variables. The grid size from the NASA POWER platform contributed to low model adjustments since the grid's extension is kilometric and can overlap areas. Despite this limitation, NASA POWER proves to be a potential tool for PMI monitoring. It should be especially helpful for growers who do not have surface weather stations near their farms.

**Keywords:** *Arachis hypogaea* L.; climate; weather data; peanut maturity (PMI); growing degree days

## 1. Introduction

Climate is extremely important in agricultural production, as a significant portion of production depends on specific climate conditions. In addition to water, temperature, relative humidity, solar radiation, and wind speed are factors that can affect production, along with the incidence of pests and diseases and soil microbiology [1]. Investigating climate change is necessary to adapt agricultural crop management, especially for plants in which growth, development, grain quality, and yield respond more sensitively to climatic variations [2].

Peanut (*Arachis hypogaea* L.) is among many crops that are affected by climate variations. Monitoring the climate has become of great importance to achieve gains in production. Peanut is produced worldwide, particularly in China (36%), India (13%), Nigeria (9%),

and the United States (USA; 5%), with the state of Georgia accounting for 44% of the USA production [3].

Given the significance of peanut production, producers have increasingly utilized technologies to aid in monitoring and decision-making for peanut cultivation. In the USA, producers have employed the PeanutFarm (http://peanutfarm.org/ accessed on 2 February 2024) system to monitor peanut plant development. This system calculates the accumulation of degree days from meteorological stations distributed in various regions [4].

Despite the system's efficiency, a network of surface meteorological stations is necessary to supply climate data. An alternative approach involves the use of surface meteorological stations in a national network [5], data series obtained from mathematical models [6,7], and the use of orbital platforms.

Installing and monitoring meteorological stations is not easy, but it is necessary to understand the climatic conditions. Nevertheless, to monitor these climatic conditions and obtain accurate and precise values, an adequate number of meteorological stations is needed. In most of the countries, the number of weather stations is adequate. The World Meteorological Organization (WMO) recommends 6.3 stations per 100 km$^2$ [5,8]. An alternative for these countries is to use data from orbital platforms (satellites) with accurate models to monitor climatic changes. For instance, the use of temperature, precipitation, and relative humidity from the orbital platform is an alternative to creating evapotranspiration models to improve irrigation management and agricultural practices [9–11].

Several countries have adopted NASA POWER for climate monitoring, providing essential climatic information. The authors in [9] used the platform to estimate evapotranspiration in Lagunera, Mexico. In the semi-arid Mediterranean, the platform showed satisfactory adjustments to estimate daily evapotranspiration and improve irrigation methods [10]. The authors in [11] evaluated the accuracy and precision of NASA POWER climatic data in different climatic zones in Egypt, comparing it with surface weather stations. In Sicily, Italy, NASA POWER was used to estimate the reference evapotranspiration and apply it in regions that did not have weather stations to understand the impact of climate changes and improve agriculture [12].

One of the main orbital platforms for climate monitoring is NASA POWER, which collects information on a $1° \times 1°$ grid for solar radiation sources and a $½° \times 5/8°$ grid for climate data, enabling global climate monitoring. This tool has been applied to estimate corn productivity [5], leaf area, and productivity in soybeans [13] and develop models for identifying thermal stress [14].

Despite the use of grid data in various crops and for different purposes, there are no reports using these data to estimate peanut pod maturity, a crucial factor in determining grain productivity and quality. The maturation process of peanut pods depends on the accumulation of degree days by plants, with high temperatures accelerating growth and development, leading to faster maturation, while low temperatures can slow growth and delay maturity and, consequently, harvest time [15]. Monitoring pod maturity is crucial for farmers to improve their production and identify the optimum timing for inverting the peanut plants. This monitoring can be done using an orbital platform, such as NASA POWER, since the platform is online and publicly available.

However, there are limitations in using grid data. The data are collected using grids, and the grids have low spatial resolution, thus resulting in a loss of quality and detail richness. Errors in climate variable measurements can be encountered, such as precipitation and wind speed [5,16], affecting data quality and leading to erroneous analyses. Nevertheless, working with orbital data-collection tools that are publicly accessible can improve the understanding of climate changes and their effects on peanut cultivation and maturation, eliminating the need for meteorological stations near production areas.

Based on this, studies are required to investigate the reliability of data obtained from orbital platforms, as well as describe which variables are reliable for use in agriculture. Therefore, the objective of this work was to verify the applicability of remotely obtained

NASA POWER data to estimate peanut pod maturity and compare the data provided by NASA POWER with data obtained from surface meteorological stations.

## 2. Materials and Methods

### 2.1. Study Location

The fields used to assess the relation between pod maturity and climate data are in Georgia, USA. The region is classified as a subtropical humid climate (Cfa: temperate, without dry season and hot summer) with annual precipitation of 1346 mm [17]. Four fields were used (Figure 1) to evaluate the peanut maturity, with two fields being irrigated (Magnolia 2018 and Docia 2019) with center pivot and the other two fields being rainfed (Blaelock 2018 and Grand Canyon 2019). At each field, georeferenced points distanced 100 m apart were inserted, with 24 points (1 point/hectare) for Blaelock, Docia, and Magnolia and 12 points (1 point/hectare) for Grand Canyon to collect maturity samples.

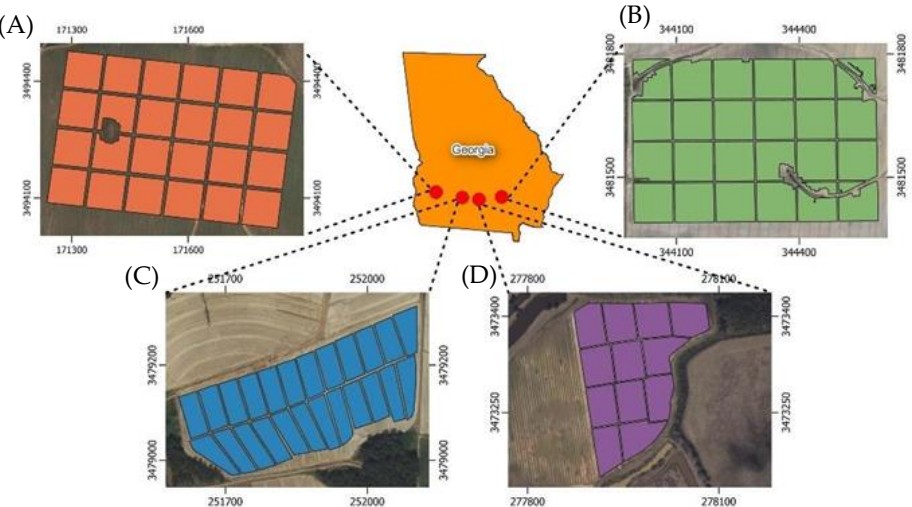

**Figure 1.** Peanut field production across various counties in Georgia, USA. (**A**) Magnolia (Ducker); (**B**) Blaelock (Coffee); (**C**) Docia (Tift); (**D**) Grand Canyon (Berrien). The red dots represent the location of each field in Georgia, and the color in each field represents the buffer delimiting the regions (polygons) of the collection of peanut to evaluate peanut pod maturity.

The fields were planted with the cultivar Georgia-06G, known for its dark green foliage and intermediate growth habit (Runner), with a production cycle of approximately 140 days [18]. Row spacing was 0.90 m. Planting was 5 June 2018 for Magnolia, 9 May 2019 for Docia, 11 June 2018, and 27 April 2019 for Grand Canyon. Each field was in a specific county: Magnolia in Ducker, Docia in Tift, Grand Canyon in Berrien, and Blaelock in Coffee.

### 2.2. Peanut Pod Maturity Evaluation

Peanut pods were collected from each field on different dates (Table 1). A total of 8 to 13 plants were collected around the georeferenced point (2 to 5 m), aiming for 200 pods per point (Figure 1). The collected plants were placed in a bag, identified, and transferred to the laboratory to detach the pods from the plants.

**Table 1.** Plant collection dates in days after sowing (DAS) from the fields for peanut pod maturity evaluation. Field counties are in parenthesis after each field name.

| Fields | DAS |
|---|---|
| Blaelock (Coffee) | 96; 104; 109; 116; 128; 135 |
| Grand Canyon (Berrien) | 103; 109; 117; 124; 131; 138; 145 |
| Docia (Tift) | 115; 122; 129; 139 |
| Magnolia (Ducker) | 96; 107; 117; 126; 135 |

Subsequently, the pods were pressure-washed, removing the exocarp and exposing the mesocarp [15], and classified according to peanut board maturity [19], which was adopted as the standard for evaluating maturity. The Peanut Maturity Index (PMI), ranging from 0 to 1, was obtained by summing up the brown and black columns of the maturation board and then dividing it by the total number of classified pods. Values close to 1 indicate mature pods, while values close to 0 indicate immature pods. However, under field conditions, optimal PMI values ranging from 0.7 to 0.75 were adopted to minimize quantitative and qualitative losses [15].

### 2.3. Climate Information

Meteorological stations located in Berrien (Alapaha), Tift (Tyty), Coffee (Douglas), and Dougherty (Ducker) were selected for collecting climatic data. These stations were the closest to the production fields described in Table 2.

**Table 2.** Distance (km) and elevation (m) of a surface weather station (SWS) of the production fields (PC) used to evaluate peanut pod maturity.

| SWS | PC | Distance (km) | Elevation (m) |
|---|---|---|---|
| Berrien | Grand Canyon | 9.0 | 82 |
| Tift | Docia | 10.5 | 113 |
| Coffee | Blaelock | 14.0 | 68 |
| Dougherty | Magnolia | 8.4 | 62 |

SWS—Surface weather stations; PC—production fields.

These stations near the collection fields for maturity assessment were selected to compose the analyses. Therefore, it was possible to carry out a comparative study between the climate variables predicted by NASA POWER and those observed by surface weather stations. The University of Georgia (UGA) Tifton Campus provided spreadsheets with weather data collected from the weather stations. However, due to the season collection system, climate variables data from Table 3 were recorded every 15 min. As the NASA POWER platform only provides daily data on climate variables, the data provided by the surface weather stations at UGA have been converted into a daily scale by calculating the average of the values of the climatic variables provided every 15 min. Thus, both stations were standard on a daily scale. In addition, data from the full years of 2018 and 2019 for the four weather stations were provided, creating the variables shown in Table 3.

**Table 3.** Agroclimatology variables obtained by fixed station and platform NASA POWER.

| Climate Data | Unit of Measurement |
|---|---|
| Wind speed [1] | m/s |
| UR [2] | % |
| Tmax [3] | °C |
| Tmean [4] | °C |
| Tmin [5] | °C |
| SWN [6] (Qg) | MJ m$^2$ dia$^{-1}$ |
| PS [7] | kPa |

[1] speed of view at 2 m high; [2] humidity relative to 2 m high; [3] maximum temperature at 2 m high; [4] average temperature at 2 m tall; [5] minimum temperature at 2 m tall; [6] surface radiation incidences (solar radiation); [7] surface pressure.

On the NASA POWER platform (https://power.larc.nasa.gov/data-access-viewer/ accessed on 2 February 2024), the geographic coordinates of each surface weather station were inserted to collect the weather information described in Table 3. The platform provides daily information for each climate variable, with information being downloaded in CSV format for the full years of 2018 and 2019 from the four weather stations (Alapaha, Tyty, Douglas, and Ducker). The spatial resolution for the platform grid was $1° \times 1°$, which

is approximately 12,347 km$^2$ for sources of primary solar radiation, whereas for weather data, regular grids of 0.5° × 0.625° of latitude/longitude, about 3850 km$^2$, were applied. The accuracy of the platform is adversely impacted by the use of large grids. For each variable, the root mean square error (RMSE) varies between 2.10, 3.15, and 3.10 °C for average, minimum, and maximum temperatures, respectively. Similarly, the RMSE values for wind speed, relative humidity, and surface pressure are 2.17 m/s, 12.06%, 2.87 kPa, and 6–12%, respectively [16,20].

The air temperature estimate was made using the Goddard Earth Observing System Global version 4 (GEOS-4), with an analysis interval of 3 h. Solar radiation data were obtained using the NASA International Satellite Cloud Climatology Project (ISCCP), and surface solar radiation was estimated using the ISCCP model [20,21].

With the collection of climate data from the two platforms, the comparison between the two forms of collection (terrestrial and orbital) was carried out, which indicated whether the NASA POWER data were accurate at estimating climate variables and consequently should be used in the estimation of peanut maturity anywhere around the globe.

### 2.4. Statistical Analysis of the Two Platforms

Climate data from the two platforms were combined into a general model, which considered the two years (2018 and 2019) of collection at all locations (Berrien, Coffee, Dougherty, and Tift). The climatic data from the two collection platforms were combined into a general model, which accounted for the two years (2018 and 2019). An additional segmentation was performed based on location, i.e., Berrien, Coffee, Dougherty, and Tift. This approach allowed for a specific analysis of each surface weather station throughout the two years of collection.

Initially, the full dataset was inserted into the exploration analysis using the boxplot, removing the values described as outliers by calculating the limits (inferior and superior). The weather data were analyzed using Pearson's correlation analysis ($p < 0.05$), and the graphs (heat maps) were plotted using the Jupyter platform with Python language. In addition to the correlation, linear regression analysis was performed for climate variables that showed a correlation coefficient above 0.8. Coefficient values between 0.67 and 1.0 [22] are generally considered to have a high correlation; however, for this study, the value of 0.8 was used to select the variables. For this analysis, the climatic variables of the surface weather stations were considered to be the dependent variables, whereas the independent variables were the variables provided by the NASA POWER platform. Exploratory analysis using boxplot and linear regression analysis was carried out using SAS© JMP pro 14 version 14.0.0 software, and regression graphs were created using Office Excel 2013 version 15.0.45 software. Subsequently, for the evaluation of the metrics of the models, the accuracy measurement of the RMSE (Equation (1)) and the determination coefficient ($R^2$) (Equation (3)) were used as a measure of precision.

Additionally, the calculation of (d), the Willmott index of conformity (1981), described in Equation (2), was carried out. The Willmott performance index (d) is a representation of the degree of error of the models, ranging from 0.0 to 1.0, with values close to 1.0 indicating a good match between the observed and predicted values [23].

$$RMSE = \sqrt{\frac{\sum_{i=1}^{n}(y_{obs} - y_{est})^2}{n}} \qquad (1)$$

$$d = 1 - \frac{\sum_{i=l}^{N}\left(Y_{obs_i} - Y_{est_i}\right)^2}{\sum_{i=l}^{N}\left(\left|Y_{est_i} - \overline{Y}\right| + \left|Y_{obs_i} - \overline{Y}\right|\right)} \qquad (2)$$

$$R^2 = \frac{\sum_{i=l}^{N}\left(Yest_i - \overline{Y}\right)^2}{\left(Yobs_i - \overline{Y}\right)^2} \qquad (3)$$

where *RMSE* is the square root of the average error, *d* is the Willmott concordance coefficient, and $R^2$ is the determination factor. $y_{obs}$ is the observed value, $y_{est}$ is the estimated value by the model, and *n* is the number of data points. $\overline{Y}$ the average value of the estimated variable.

Following the linear regression analysis, it was possible to show whether the NASA POWER platform is accurate and precise for estimating the climate variables found in Table 3. The variables were inserted in the principal component analysis (PCA). Thus, the dataset used was restricted to the periods of evaluation of maturity in the fields, and the PCA was carried out for each field (Berrien, Dougherty, Tift, and Coffee) and the Global model. Ultimately, the relationship between the PMI and climate variables can be identified, and one can select those variables that show the best results. With the PCA, the variables that best correlate with PMI can be selected, and the variables not showing a strong relationship can be excluded. PCA reduces the dimensionality of data while retaining as much information as possible. By transforming the data into principal components, it becomes feasible to concentrate on the directions that encompass the highest variability, therefore eliminating redundancies and emphasizing significant patterns. This technique is particularly valuable when dealing with datasets featuring numerous variables, aiding in the simplification of analysis and interpretation. The PCA was carried out using the software R, version 2023.06.2, and the package "factoextra", as shown in Figure 2.

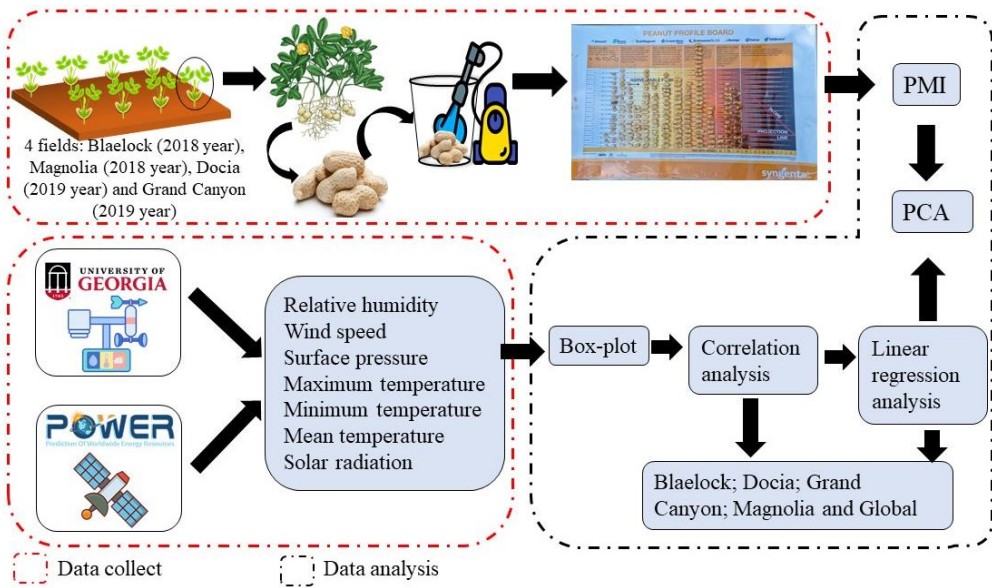

**Figure 2.** Evaluation steps for maturity classification and comparison between the NASA POWER platform and surface weather stations.

## 3. Results and Discussion

### 3.1. Correlation Analysis

The lowest correlation coefficients between climate data-collection platforms were found for wind speed (WS) and relative humidity (UR), 0.58 and 0.61 in the Global model, respectively (Figure 3E). For surface pressure (PS), solar radiation (Qg), maximum temperature, minimum temperature, and mean temperature, coefficient values were higher than 0.84 for the Global model (Figure 3E). The weak correlations for the UR and WS variables were due to interference in their localization, topography, and change in land use, which can cause errors in measurements when using grid data [11,16]. On the other hand, despite having a correlation value of 0.84, the PS perceived low reliability over weather stations in other studies [9], which also assessed the efficiency of NASA POWER.

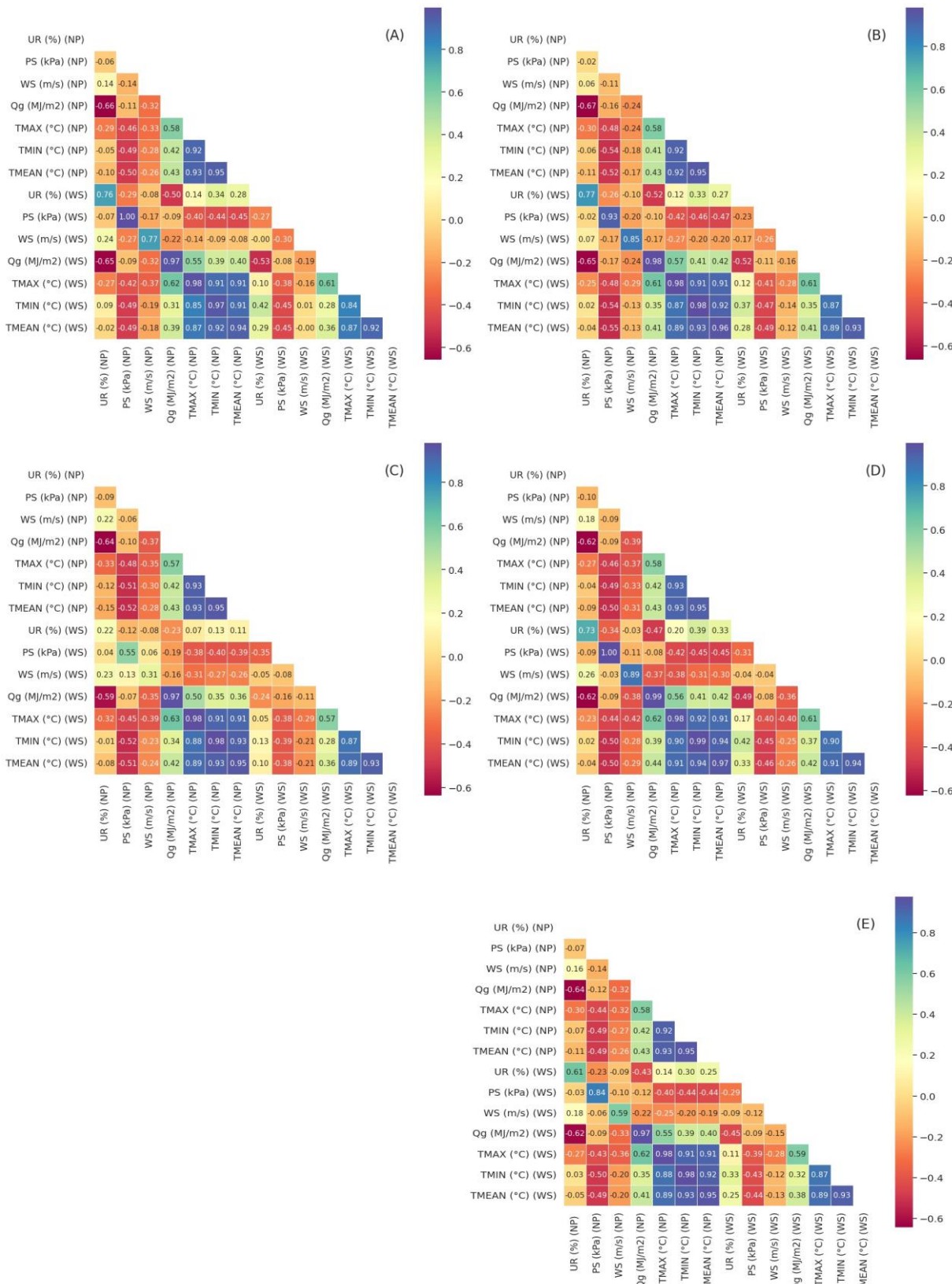

**Figure 3.** Correlation analysis for multiple variables from the locations Berrien (**A**), Coffe (**B**), Dougherty (**C**), Tift (**D**), and general model (**E**) for the two years (2018 and 2019).

When the correlation analysis was performed by location for Berrien, Coffee, and Tift (Figure 3A,B,D), the coefficients for the variables UR, WS, and PS were higher. Among these variables, WS changed from 0.59 in the global correlation (Figure 3E) to 0.85 and 0.89 for Coffee (Figure 3B) and Tift (Figure 3D), respectively. Although correlation values for WS had improved for these locations, when these variables (UR, WS, and PS) were analyzed individually in Dougherty (Figure 3C), the correlation values were lower. For maximum, minimum, and average temperatures and solar radiation, all locations showed similar results to the overall model.

The increase in the correlation coefficient can be attributed to the location conditions of each surface weather station since, in the NASA POWER data collection, there was no overlap of grids. Despite this, topography is one of the main factors affecting this relationship. The greater the elevation, the greater the errors described by the NASA POWER platform [12,24,25].

When the data were collectively analyzed to create a general model, the Pearson correlation values among all climatic variables showed a decrease (Figure 3E), i.e., Dougherty was the location that presented the lowest values for the Pearson correlation, with the greatest difference between the two platforms. When the locations were combined (Global), the correlation coefficient for the Global model decreased.

*3.2. Regression Analysis*

After conducting the correlation analysis, linear regression was performed using the climate variables, excluding relative humidity due to its Pearson correlation values being less than 0.8. The dependent variables (Y) were the surface weather station data, and the independent variable (X) was the NASA POWER data. Linear regression analysis was used to analyze the response of the variables and create a model to evaluate the precision ($R^2$) and accuracy (RMSE).

The linear regression analysis demonstrated that the precision values ($R^2$) for variable surface pressure (PS) were high for Berrien and Tift (Figure 4d,c), being $R^2$ = 0.99 and RMSE = 0.04 kPa for both locations. On the other hand, for the general model and the individual model for Coffee (Figure 4a,b), RMSE values were 0.25 and 0.18 kPa, and $R^2$ were 0.71 and 0.86, respectively. The results for Dougherty (Figure 4e) showed the largest variations between platforms for PS, which resulted in low $R^2$ (0.30) and high RMSE (0.36 kPa). In addition, the Willmott performance index (d) showed no variation in any of the analyses and local data for surface pressure.

Surface pressure was related to the displacement of water in the soil, causing the process of absorption of water and nutrients by the roots of plants. In addition, surface pressure is related to site topography, with regions with higher elevations showing lower PS, while higher PS values were observed for regions with lower elevations [25].

The WS had the highest variations in accuracy and precision levels in different locations. Dougherty (Figure 5e) was the county that showed the lowest adjustments of $R^2$ (0.09) and high RMSE (0.62 m/s), followed by the overall model (Figure 5a) with $R^2$ and RMSE values of 0.34 and 0.65 m/s, respectively. When there was separation by counties, the linear regression models for Berrien, Coffee, and Tift (Figure 5d,b,c) showed a greater adjustment of $R^2$, ranging from 0.59 to 0.79, and lower RMSE, from 0.36 to 0.42 m s$^{-1}$.

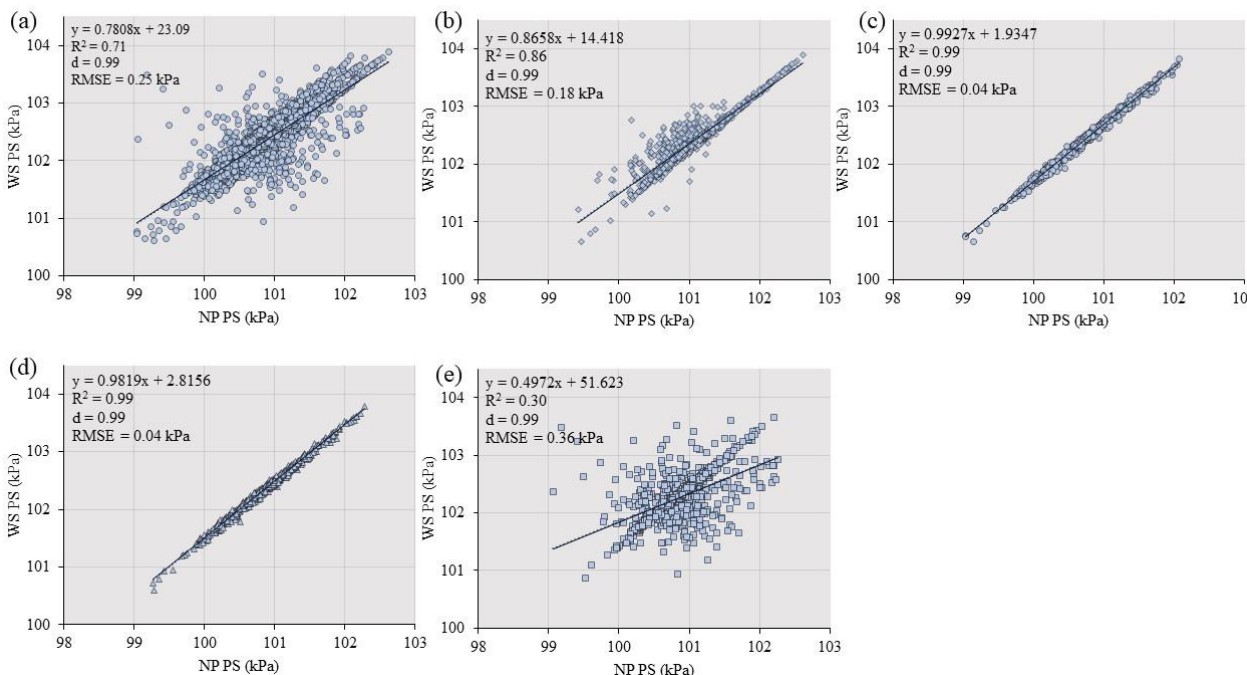

**Figure 4.** Linear regression analysis between NASA POWER (NP) and weather stations (WS) and metrics to evaluate the performance: determination coefficient ($R^2$), Root Mean Square Error (RMSE), and Willmott performance index (d) for surface pressure (PS). (**a**) General model; (**b**) Model for the Coffee region; (**c**) Model for the Tift region; (**d**) Model for the Berrien region; and (**e**) Model for the Dougherty region.

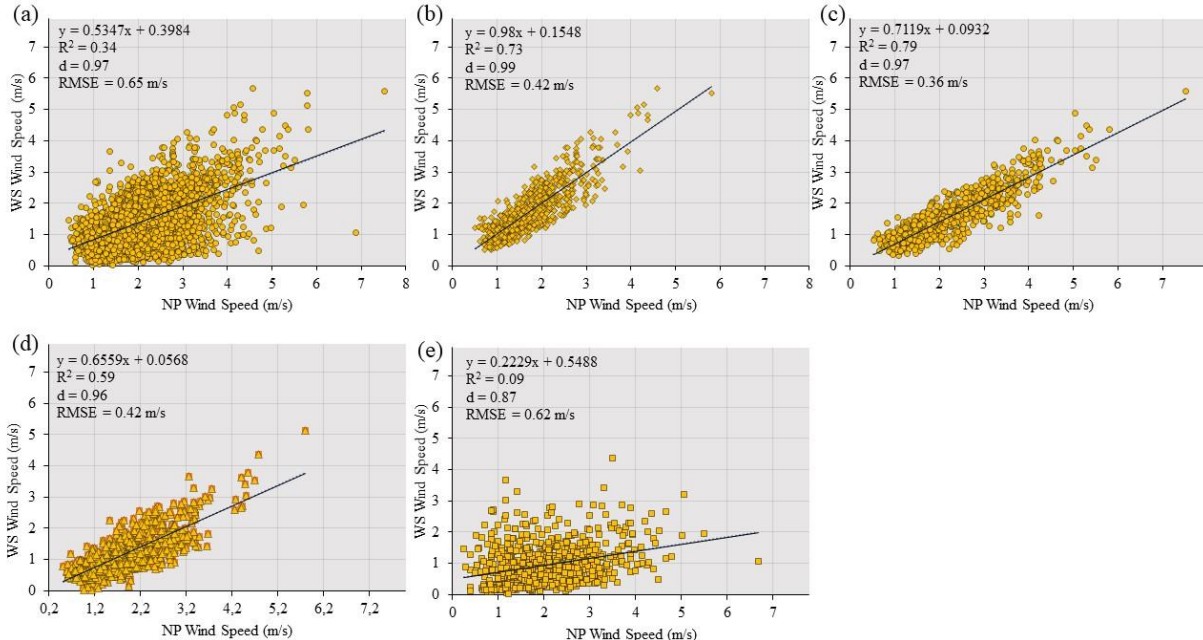

**Figure 5.** Linear regression analysis between NASA POWER (NP) and weather stations (WS) and metrics to evaluate the performance: determination coefficient ($R^2$), Root Mean Square Error (RMSE), and Willmott performance index (d) for Wind speed (WS). (**a**) General model; (**b**) Model for the Coffee region; (**c**) Model for the Tift region; (**d**) Model for the Berrien region; and (**e**) Model for the Dougherty region.

Despite the variations observed in the $R^2$ and RMSE values for the different models, in comparative studies between the NASA POWER platform and the national network

stations in Brazil, $R^2$ values ranging from 0.09 to 0.16 and RMSE from 0.93 to 1.82 m s$^{-1}$ were observed [5]. As reported by [9], models had RMSE values ranging from 0.92 to 1.63 m s$^{-1}$ and $R^2$ between 0.19 to 0.52, which were close to the variations observed in this study. In addition, for most locations, the data points shown in Figure 5a–e are more concentrated around 1.2 to 3.2 m s$^{-1}$, being more dispersed outside this range.

The variability observed in the measurement of wind speed (WS) is associated with how these data are captured by the sensor and calculated through mathematical models. The mathematical models used in the NASA POWER platform are the Modern-Era Retrospective Analysis for Research and Applications 2 (MERRA-2), which calculates the speed and direction of the wind, and the results are compared with NASA's weather stations, with RMSE values of up to 2.47 m s$^{-1}$ [20].

In agriculture, wind speed is an important factor for crop evaporation. In defining the planting window for peanut crops, plantings occurring in mid-May showed greater evapotranspiration values that increased leaf area [26]. This fact, which coincides with the season of the highest wind speed values (Figure S2), was recorded in the spring season at the beginning of sowing in Georgia, USA.

All models for daily solar radiation (Qg) showed $R^2$ adjustments above 0.94. Higher $R^2$ and lower RMSE values were observed for the general model and Berrien (Figure 6a,d). In Tift, the best adjustments for Qg were observed with $R^2 = 0.97$ and RMSE = 1.12 MJ m$^2$/day (Figure 6c). The variation in RMSE values was 0.48 MJ m$^2$/day between the highest (Figure 6a) and the lowest (Figure 6c) values observed, and there were no variations for any model in the Willmott performance index values (d = 0.99). The errors found in this study can be considered to be low since the error values for Qg estimated by NASA POWER range from 2.73 to 3.41 MJ m$^2$/day [5,9].

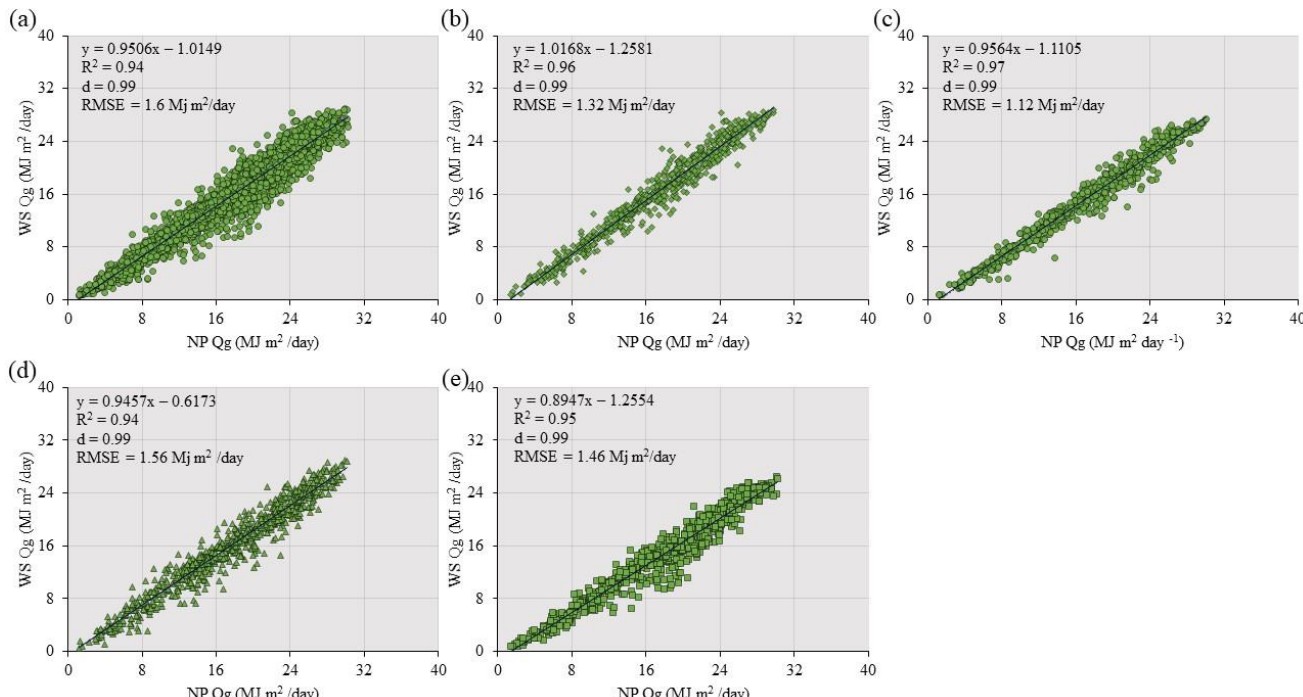

**Figure 6.** Linear regression analysis between NASA POWER (NP) and weather stations (WS) and metrics to evaluate the performance: determination coefficient ($R^2$), Root Mean Square Error (RMSE), and Willmott performance index (d) for Solar radiation (Qg). (**a**) General model; (**b**) Model for the Coffee region; (**c**) Model for the Tift region; (**d**) Model for the Berrien region; and (**e**) Model for the Dougherty region.

Solar radiation is a parameter dependent on weather conditions, and the presence of clouds makes its analysis process more challenging, leading to errors [5] and, therefore,

decreasing accuracy and precision. However, the models showed satisfactory values of accuracy and precision, even when data from all sites from the two years (Global) and when separated by location were used, demonstrating the platform's precision in estimating Qg at any time of the year. For the calculation of solar radiation, the mathematical model Global Energy and Water Exchanges (GEWEX) used by the NASA POWER platform features more satellites that capture information about cloud coverage, as well as other satellites to provide the temperature and gas data in the atmosphere. These satellites provide information to the radiative transfer models for the correction of the effects of these constituents on the estimation of solar radiation [20].

Furthermore, Qg is a temperature-dependent parameter that can influence both air and soil humidity. The seasons with the highest mean temperature and relative humidity (see Supplementary Materials)—spring and summer—recorded the highest Qg values. This period of elevated Qg corresponds to the peanut-growing season in Georgia. Therefore, late-summer seedings outside the optimal planting window may result in reduced productivity due to changes in climate conditions, particularly in temperature and solar radiation. The decrease in solar radiation and temperature decreases leaf photosynthetic rates, leading to a reduction in plant growth, biomass accumulation, and decreased productivity [26].

For maximum temperature, the observed $R^2$ values ranged from 0.95 for the general model, Coffee and Tift (Figure 7a–c) to 0.96 for Berrien and Dougherty (Figure 7d,e). The lowest RMSE values were 1.53 °C for Berrien (Figure 7d), and the highest of 1.63 °C (Figure 7e) was for Dougherty, with a variation of 0.1 °C between maximum and minimum RMSE observed in these two locations. The Willmott performance index (d) was 0.99, showing no significant difference for any of the tested models and for maximum, mean, and minimum temperatures.

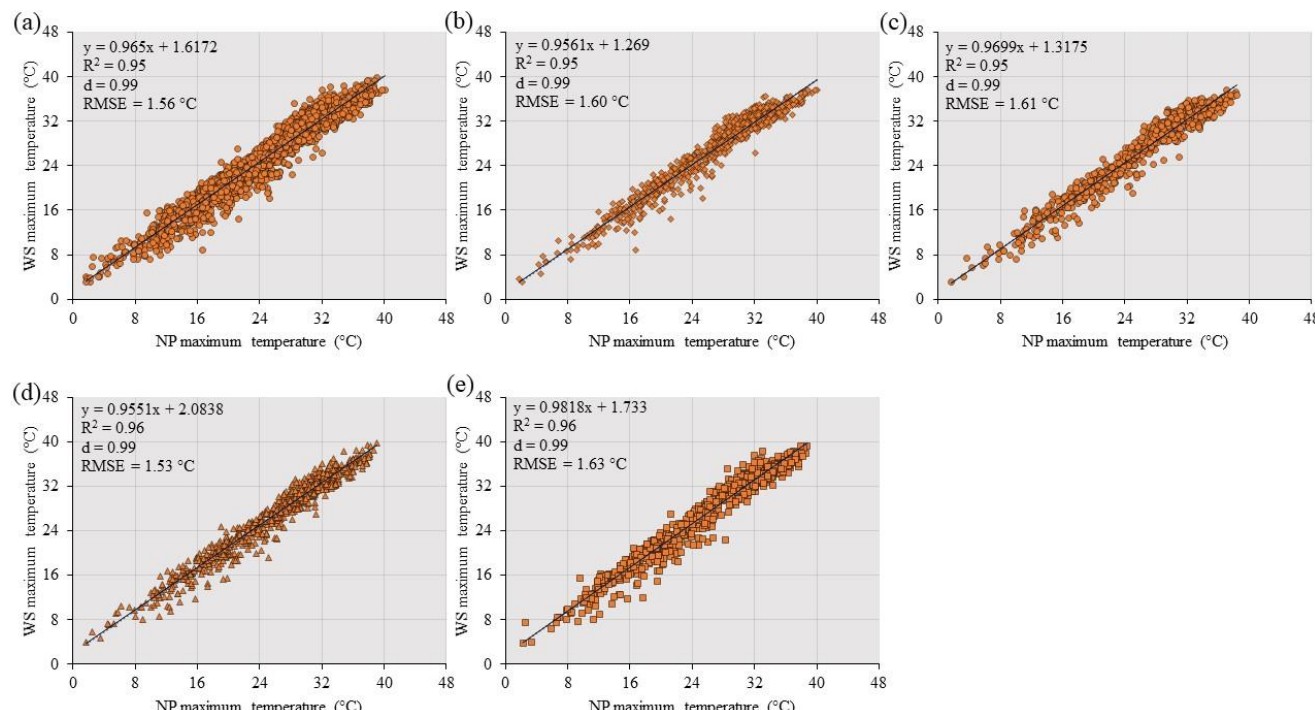

**Figure 7.** Linear regression analysis between NASA POWER (NP) and weather stations (WS) and metrics to evaluate the performance: determination coefficient ($R^2$), Root Mean Square Error (RMSE), and Willmott performance index (d) for maximum temperature. (**a**) General model; (**b**) Model for the Coffee region; (**c**) Model for the Tift region; (**d**) Model for the Berrien region; and (**e**) Model for the Dougherty region.

Both data-collection platforms recorded negative values for the minimum temperature variable. Despite this, satisfactory adjustments were obtained from the models described by

the precision values ($R^2$) of 0.96 for the general model, Coffee and Dougherty (Figure 8a,b,e), 0.97 for Tift (Figure 8c), and 0.94 for Berrien (Figure 8d). With regard to accuracy, Tift was the region with the lowest values of 1.24 °C (RMSE), whereas Berrien was the one with the highest levels of 1.84 °C (Figure 8c,d).

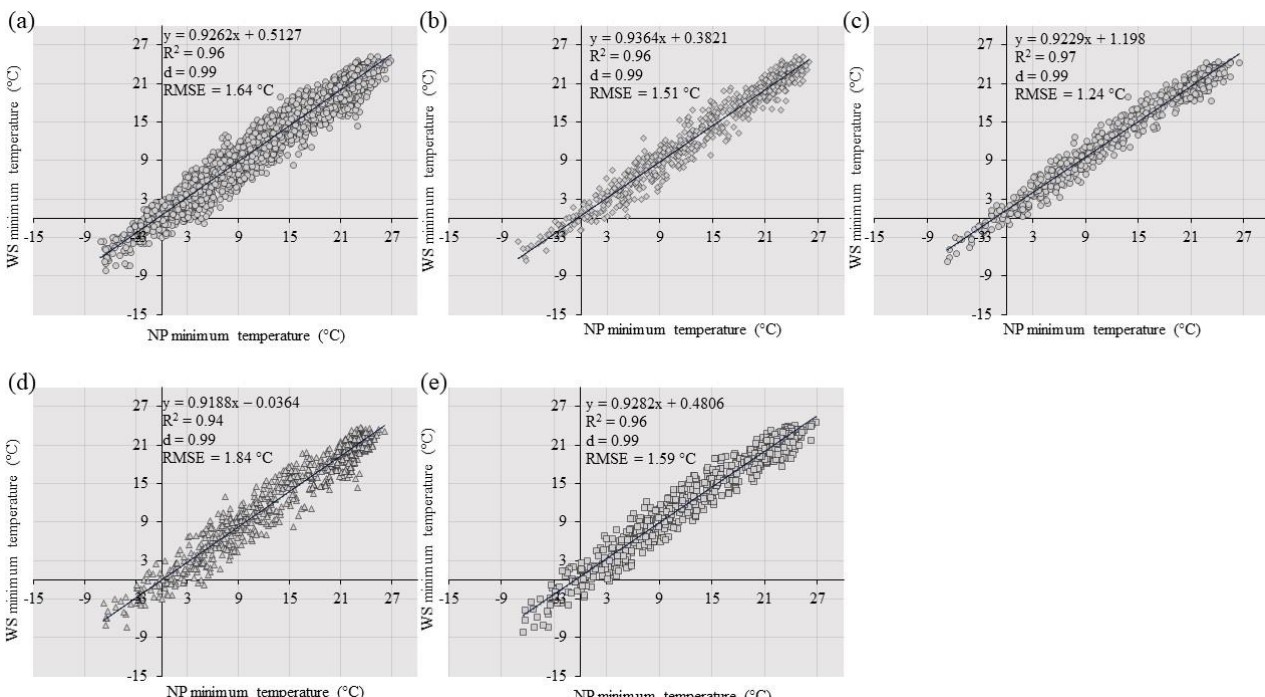

**Figure 8.** Linear regression analysis between NASA POWER (NP) and weather stations (WS) and metrics to evaluate the performance: determination coefficient ($R^2$), Root Mean Square Error (RMSE), and Willmott performance index (d) for minimum temperature. (**a**) General model; (**b**) Model for the Coffee region; (**c**) Model for the Tift region; (**d**) Model for the Berrien region; and (**e**) Model for the Dougherty region.

The linear regression models exhibited the least adjustments for average air temperature. In the general model, Coffee and Dougherty (Figure 9a,b,e) demonstrated an accuracy value of $R^2 = 0.91$, while Tift exhibited an $R^2 = 0.93$ (Figure 9c). Conversely, Berrien (Figure 9d) displayed the lowest $R^2 = 0.89$. The RMSE varied from 2.44 °C for Tift (Figure 9c) to 3.43 °C for Berrien (Figure 9d).

For maximum and minimum air temperatures, low variations in the data distribution in the regression line were recorded, resulting in satisfactory adjustments for the linear regression models. On the other hand, for the average air temperature, data points were more scattered from the adjustment line of the linear regression model. Such data dispersion can be attributed to the way data are collected from the two platforms. Although one platform recorded a positive value, the other platform recorded a negative value for the same date, affecting the fit of the models and, consequently, the parameters of accuracy (RMSE) and precision ($R^2$). The dispersion error persists due to the computational approach of the NASA POWER platform models. Given that the data are presented in grids ($0.5° \times 0.625°$) by the MERRA-2 model, the substantial extent of the grid, exceeding 50 km, may introduce errors in calculations. This discrepancy is particularly notable in locations where weather conditions diverge from those observed by nearby field weather stations. Unlike weather stations, which gather more precise information from specific locations, the grid-based approach may aggregate diverse conditions, contributing to inaccuracies.

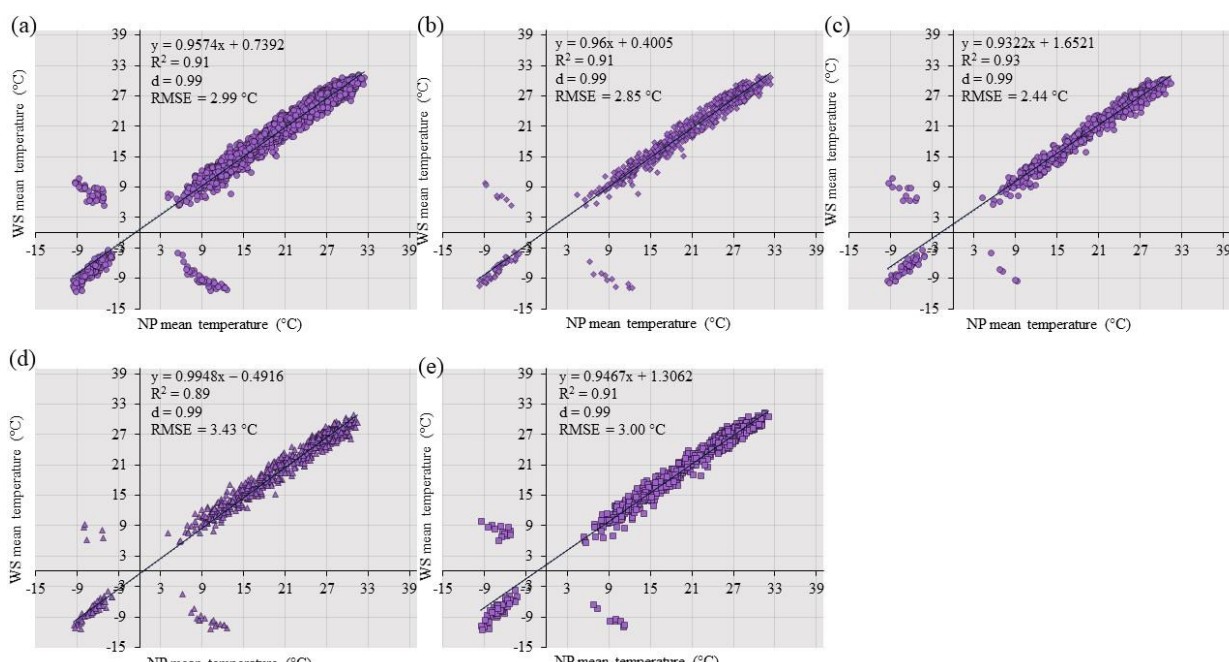

**Figure 9.** Linear regression analysis between NASA POWER (NP) and weather stations (WS) and metrics to evaluate the performance: determination coefficient ($R^2$), Root Mean Square Error (RMSE), and Willmott performance index (d) for mean temperature. (**a**) General model; (**b**) Model for the Coffee region; (**c**) Model for the Tift region; (**d**) Model for the Berrien region; and (**e**) Model for the Dougherty region.

In studies with the NASA POWER platform and surface ground stations, $R^2$ values ranging from 0.84 to 0.95 and RMSE from 1.29 °C to 3.67 °C for maximum, minimum, and average air temperature were observed [9].

In different scenarios, while working with a network of stations in Brazil, the RMSE can range from 2.64 °C to 2.83 °C, with corresponding $R^2$ values varying between 0.68 and 0.65 for mean temperature [5]. Similarly, for maximum temperature, the $R^2$ values range from 0.08 to 0.63, and for minimum temperature, they vary from 0.08 to 0.85 [27]. On the other hand, the NASA POWER program reported that errors (RMSE) of 2.10 °C were found in models (MERRA-2) when estimating average air temperature [20,21]. Conversely, in this study, great adjustments were observed for the models, described by high values of $R^2$ and low RMSE for maximum and minimum air temperatures. The topographic conditions and soil usage are crucial factors for characterizing the climate of a site. In Georgia, varying temperature ranges have been recorded, influenced by the region within the state. For northern regions near Tennessee, temperature ranged from 3.0 to 5.9 °C in January, while in the southern regions near Florida, temperature variations were from 8.0 to 14.9 °C for the same month of the year [28,29].

Determining the optimal sowing timing for crops is crucial to securing favorable climate conditions for cultivation. Depending on the seeding season, temperature significantly impacts dry-matter production, leaf growth, and peanut germination. However, the rapid initial growth, influenced by the elevated temperatures in June (27 and 33 °C), plays a significant role in plant-stand establishment and, consequently, production [30].

The peanut plant is substantially impacted by temperature, which can affect both the maturing process and the overall quality of harvested pods. This becomes apparent when employing agrometeorological indices, such as accumulated degrees days (AGD), for evaluating PMI. This index has already been used in various studies since the responses observed between maturity and AGD are satisfactory [4,15,31]. For the calculation, in addition to maximum and minimum air temperatures, the base temperature of 13.5 °C for

peanuts is required [26,30]. Another important aspect is that temperature can also influence other climate variables, therefore altering how the crop is managed.

Temperature is an important factor from a climate point of view, and climate variables such as relative humidity and solar radiation are influenced by their changes. In the management of agricultural crops, these climate variables collaborate to obtain high productivity. However, for the monitoring of weather conditions, grid platforms such as NASA POWER are low-cost and feasible tools that can be applied to the analysis of local weather conditions. Such platforms show similar results when compared to surface weather stations, mainly for the variables of surface pressure, maximum, minimum, and average air temperature, and solar radiation (Figures 4–9).

### 3.3. Principal Component Analysis (PCA)

Climate data are relevant from an agronomic point of view, as they can interfere with the productivity of agricultural crops. However, to assess the climatological variables that are most related to maturity in peanut, the principal component analysis (PCA) shown in Figure 10 was performed. In the PCA analysis, the variables surface pressure, relative humidity, maximum, minimum, and average temperatures, as well as wind speed and solar radiation, were included, and the relationship of climate variables with maturity (PMI) could be observed.

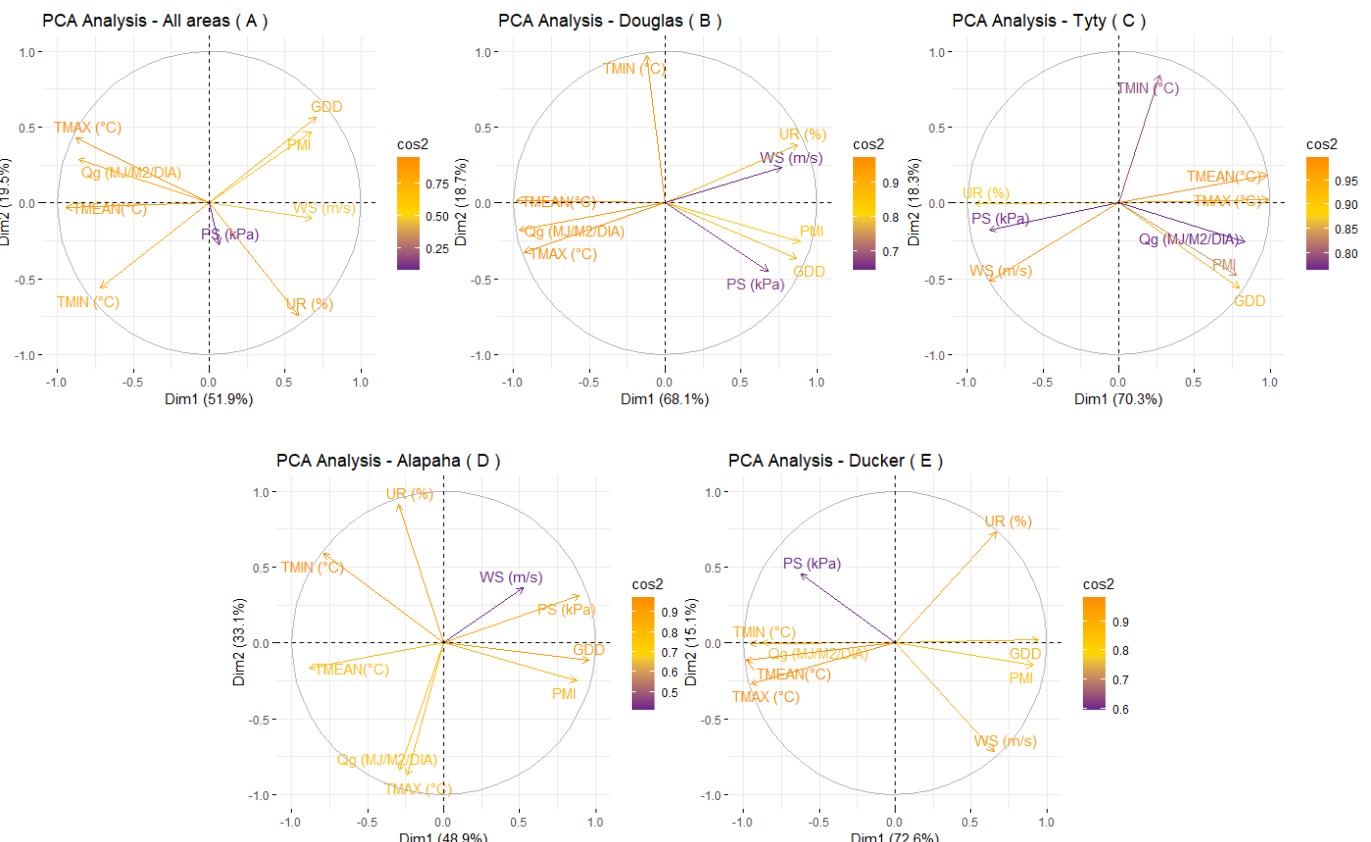

**Figure 10.** Principal Component Analysis (PCA) for each region and Global model. (**A**) represents the PCA of the general model; (**B**) represents the Coffee region PCA; (**C**) represents the Tift region PCA; (**D**) represents the Berrien region; and (**E**) represents the Dougherty region. PS: surface pressure; WS: wind speed; UR: relative humidity; QG: solar radiation; Tmax: maximum temperature; Tmin: minimum temperature; Tmean: average temperature.

The general model (Figure 10A), considering the two PCA components, accounted for 71.8% of the data variability. For the analysis by region, the sum of components 1 and 2 accounted for 82%, 87,1%, 87.7%, and 88.2% of the overall data variability for

Berrien (Figure 10D), Coffee (Figure 10B), Dougherty (Figure 10E), and Tift (Figure 10C), respectively. These results demonstrated an increase in PCA's ability to respond to data variability. Furthermore, it was observed that Tmin, Tmax, Tmean, and Qg follow an opposite trend than PMI, demonstrating that such climate variables affect PMI values, i.e., as temperature or solar radiation increase, PMI value decreases, consequently decreasing the productivity and quality of harvested pods.

Thus, the definition of the growing season is an essential factor for the development of peanut plants. The sowing window for peanuts begins around 10 April and lasts until the beginning of July. In this seeding season, the ideal climatic conditions for peanut are found, with the optimal temperature for growth being 27.5 °C, which can vary from 29 °C to 33 °C [31]. These variables that may interfere with the development of peanut plants can be estimated (maximum, minimum, average air temperatures, and solar radiation) using the NASA POWER platform (Figures 6–9) with high accuracy and precision. However, it should be observed whether peanut-growing areas do not overlap due to the low spatial resolution of 0.5° × 0.625° of latitude and longitude (approximately 55.6 × 69.4 km) for meteorological data and 1° × 1° of latitude and longitude for solar radiation data (approximately 111 × 111 km) of grid data.

## 4. Conclusions

In conclusion, this study has successfully demonstrated the viability of utilizing NASA POWER data for monitoring climatic conditions, showcasing strong correlations between maximum, minimum, and average air temperatures, as well as solar radiation when compared to surface weather stations. Notably, these variables exhibited significant relationships with peanut pod maturity, as highlighted in the PCA analysis.

Despite the promising results, certain limitations were identified, particularly for wind speed, which displayed challenges in achieving accurate and precise adjustments in linear regression models. This discrepancy can be attributed to the difference in measurement heights between NASA POWER (50 m) and weather stations (2 m), impacting the overall fit, especially in the Dougherty region.

However, the NASA POWER platform emerges as a valuable tool for climatic monitoring. Farmers can leverage this platform to gain insights into crop behavior across diverse climates. The broader application extends to areas without surface weather stations, enabling accurate monitoring and providing a useful tool for understanding climatic changes.

Although the platform proves instrumental, it is essential to acknowledge its limitations, such as low spatial resolution with grids larger than 50 km for weather data and more than 100 km for solar radiation. This may introduce restrictions and potential interference in data analysis. For instance, relative humidity showed a correlation below 0.8 when compared to surface weather stations, suggesting caution in its interpretation, especially in regions with monitoring stations reporting errors.

In terms of innovation, this work proposes a groundbreaking approach to monitoring climatic conditions on farms using publicly accessible orbital platforms. The NASA POWER platform stands out as an excellent resource, empowering farmers in peanut fields to calculate indices and effectively monitor climate parameters. Moreover, regions lacking surface weather stations can rely on the orbital platform to access crucial climatic information, contributing to more informed agricultural practices and climate monitoring on a broader scale.

**Supplementary Materials:** The following supporting information can be downloaded at https://www.mdpi.com/article/10.3390/agriengineering6010027/s1. Display of boxplot graphs illustrating the variability throughout the seasons of the year between the NASA POWER platform and surface weather stations.

**Author Contributions:** For the research article, the authors' contributions are as follows: Conceptualization, C.P., A.f.d.S. and G.V.; Methodology A.F.d.S. and T.T.B.V.; Validation T.O.C.B. and M.A.J.F.; Formal Analysis A.F.d.S., T.O.C.B. and M.A.J.F.; Writing—Review and Editing T.O.C.B., A.F.d.S. and C.P.; Visualization T.T.B.V. and G.V.; Project Administration A.F.d.S.; Writing—original draft preparation T.O.C.B. All authors have read and agreed to the published version of the manuscript.

**Funding:** This research was funded by the Minas Gerais State Research Support Foundation (FAPEMIG), process number APQ-00219-21.

**Data Availability Statement:** Data are contained within the article.

**Acknowledgments:** The authors acknowledge that the Minas Gerais State Research Support Foundation (FAPEMIG) funded the project and the Coordination of Superior Level Staff Improvement (CAPES) by the financial support of students. The Federal University of Lavras (UFLA) made the laboratory available and contributed to the structure. The University of Georgia (UGA) helped during the analysis, revisions, and collaboration with the data.

**Conflicts of Interest:** The authors declare that the research was conducted in the absence of any commercial or financial relationships that could be construed as a potential conflict of interest.

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
