# Peer review of "Advanced Farming Strategies Using NASA POWER Data in Peanut-Producing Regions without Surface Meteorological Stations"

_agriengineering, doi:10.3390/agriengineering6010027_

Round 1

Reviewer 1 Report

Comments and Suggestions for Authors

The paper titled: “Advanced Farming Strategies Using NASA-POWER Data in Peanut-producing Regions without Surface Meteorological Stations” submitted by the authors Barboza et al investigated  the application of remotely obtained NASA-POWER data to estimate peanut pod maturity and compare the data provided by NASA-POWER with data obtained from surface meteorological stations.

The study is interesting for me.

There are some points need to be addressed before the publishing of this paper:

1.       The introduction

-           Is well written .

2.       The material and methods : 

-          Line 83 , please insert the climate conditions or describe it as well as the citation as used  

-           Line 101 the sentence need to be rephrased, actually the whole paragraph need to be rephrased or rewritten, for better understanding.

-          Line 105, table 1 title is not clear please replace the word epochs? Its not explained before in the materials and methods.

-          Please add reference to the methodology described in lines 165-176.

-          Figure 1, add description to the cites or remove it

3.       Results and discussion

In general, the results and discussion are well presented, however some minor points need to be corrected such as :

-          Figure 2C is blurry and need to be replaced.

4.       The conclusion is missing for the prospect of the work, please add it.  

I give you Major revision.

Author Response

#Review 1:

The paper titled: “Advanced Farming Strategies Using NASA-POWER Data in Peanut-producing Regions without Surface Meteorological Stations” submitted by the authors Barboza et al investigated the application of remotely obtained NASA-POWER data to estimate peanut pod maturity and compare the data provided by NASA-POWER with data obtained from surface meteorological stations.

The study is interesting for me.

We are glad to hear that the paper caught your interest! Thank you in advanced for your comments to improve our paper.

There are some points need to be addressed before the publishing of this paper:

  1. The introduction

Thank you for your positive feedback

  1. The material and methods:

-   Line 83, please insert the climate conditions or describe it as well as the citation as used 

In accordance with the Koppen and Geiger classification, Line 83 now encompasses a detailed description of the climate conditions. The study area is characterized by a subtropical humid climate featuring a hot summer, annual precipitation measuring 1346 mm. Additionally, the mean temperature during the coldest season hovers around 0 to -3°C. Moreover, the temperature profile includes a notable aspect—specifically, one month with a mean temperature surpassing 22°C, accompanied by a substantial span of four months where the mean temperature remains above 10°C. The citation for this classification has also been incorporated, ensuring transparency and accuracy in referencing.

- Line 101 the sentence needs to be rephrased, actually the whole paragraph need to be rephrased or rewritten, for better understanding.

 The sentence at Line 101 has been revised for better clarity. In this phase of the study, we detail the process of plant collection in each field (Blaelock, Grand Canyon, Docia, Magnolia), commencing on the days specified in Table 1 as days after sowing (DAS). Plants were gathered within a georeferenced range of 2 to 5 meters from each point. Subsequently, the collected plants were placed in bags, identified, and transported to the laboratory. In the lab, the pods were meticulously removed, aiming for 200 pods per collection point.

Notably, the initial collections involved a higher number of plants (13), gradually decreasing to 8 towards the conclusion of the collection period. This variance is attributed to the lower pod count at the start of the collection, while towards the end, the plants exhibited a higher pod yield, rendering the collection of additional plants unnecessary

-  Line 105, table 1 title is not clear please replace the word epochs? It’s not explained before in the materials and methods.

 The term 'epochs' in the table title at Line 105 has been replaced with 'date' for improved clarity. The revised title now explicitly describes the dates of plant collection in each field. For instance, in the Blaelock field, plant collection commenced 96 days after sowing, providing a more transparent and accessible understanding.

- Please add reference to the methodology described in lines 165-176.

 In this section, we applied a methodology based on authors experience about peanut maturity data, incorporating data exploration and outlier exclusion using tools such as the box plot. Subsequently, we calculated the Pearson correlation coefficient and generated a heatmap for variable analysis. To determine variables for the next step, which involves linear regression models, we selected correlation coefficient values above 0.8. It's important to note that the methodology employed in this section was created by the authors specifically for investigating the data.

- Figure 1, add description to the cites or remove it

We insert the counties for each field.

“Figure 1. Peanut fields' production across various counties in the state of Georgia, USA. A)Magnolia (Ducker); B) Blaelock (Coffee); C) Docia (Tifton); D) Grand Canyon (Berrien).”

  1. Results and discussion

In general, the results and discussion are well presented, however, some minor points need to be corrected such as:

Thank you for your positive feedback on the overall presentation of the results and discussion. I appreciate your guidance.

-  Figure 2C is blurry and need to be replaced.

The Figure 2C was change and corrected the quality of the image.

  1. The conclusion is missing for the prospect of the work, please add it.

 We have revised the content to incorporate this perspective. Your insight is greatly appreciated.

 I give you Major revision.

Thank you for your feedback. I have carefully addressed all the reviewer comments in the revision. I would appreciate any specific guidance or further suggestions you may have to ensure the manuscript meets the desired standard. I am committed to making any necessary revisions to improve the overall quality of the paper.

Reviewer 2 Report

Comments and Suggestions for Authors

This study “Advanced Farming Strategies Using NASA-POWER Data in Peanut-producing Regions without Surface Meteorological Stations” aimed to find out the impact of the climate variables associated with peanut maturity, as well to compare data sourced from the NASA-POWER platform with information from surface weather stations.

The study showed that the maximum, minimum, average air temperature and solar radiation, were the ones that most showed relationship with maturity of peanut, On the other hand, they were the variables in which NASA-POWER data showed good relationships with surface meteorological stations.

Specific comments:

Table 2: Change “9” for “9,0” and change “14” for “14,0”

Table 3: In the note at the end of the table you have: “3 maximum temperature at 2 meters high; 4 average temperature at 3 meters tall; 5 minimum temperature at 5 meters tall. Are they different sensors for fixed stations?

Author Response

#Review 2:

This study “Advanced Farming Strategies Using NASA-POWER Data in Peanut-producing Regions without Surface Meteorological Stations” aimed to find out the impact of the climate variables associated with peanut maturity, as well to compare data sourced from the NASA-POWER platform with information from surface weather stations.

The study showed that the maximum, minimum, average air temperature and solar radiation, were the ones that most showed relationship with maturity of peanut, On the other hand, they were the variables in which NASA-POWER data showed good relationships with surface meteorological stations.

We sincerely appreciate the valuable insights provided by the reviewer. Their constructive comments have been instrumental in enhancing the quality and clarity of our work. We are grateful for the time and effort dedicated to the review process.

Specific comments:

Table 2: Change “9” for “9,0” and change “14” for “14,0”

We changed the numbers in the table.

Table 3: In the note at the end of the table you have: “3 maximum temperature at 2 meters high; 4 average temperature at 3 meters tall; 5 minimum temperature at 5 meters tall. Are they different sensors for fixed stations?

 We appreciate the keen observation. The numbers in the note at the end of Table 3 have been rectified for accuracy. It was indeed an oversight in the original placement of the numbers. For instance, in the sentence '4 average temperature at 3 meters tall,' the correct information is '4 average temperature at 2 meters tall.' We have ensured that all sensors, including those for temperature, wind speed, and relative humidity, consistently collect data at the 2-meter height, and the appropriate adjustments have been made throughout the text to reflect this correction.

Reviewer 3 Report

Comments and Suggestions for Authors

The article deals with a very interesting topic regarding the ‘Advanced Farming Strategies Using NASA-POWER Data in Peanut-producing Regions without Surface Meteorological Stations’. Overall, it is a comprehensive article and the findings provided indicate that a great deal of effort was put in. Suggestions for improvements that could be performed to the manuscript prior to its publication are the following:

Abstract:

-It is suggested to avoid acronyms, numerical data and parentheses in this part of the manuscript. Try to keep the Abstract simple yet informative and focus on the research questions that the authors try to answer through the paper, as well as the novelties/results that this paper presents. In addition, the content through lines 30-31 needs to be rephrased.

Introduction:

-It would be interesting to add some corresponding references about the situation in other countries and resume what has been done so far in this field. In addition, it is suggested to add some more comments regarding the novelties that this paper presents. What research gaps does this paper try to fill up regarding previous similar studies?

- Regarding the citation of your references: Try to use the method proposed by the journal throughout the whole manuscript (keep the same style and make sure you follow a uniform pattern, see lines 84-85 for instance).

Materials and Methods

-Line 83 Typo: 'Figura' -> Figure (please check all captions throughout the manuscript as well and make sure you follow a uniform pattern). Check spelling and English grammar.

-Lines 142-149: Any comments about the accuracy of the data collected (given by the providing platform maybe)?

-Line 197: It would be useful to your readers to elaborate more on PCA analysis (the same goes for regression analysis -line 244)

- It is suggested to add a general workflow showing what steps were made, in order to facilitate reading.

Conclusions

- Lines 459-460: It is suggested to rephrase the sentence.

- It would be interesting to comment on the following: What could be some possible restrictions or challenges if the proposed methods were implemented in a different area of interest? Are there any other factors that should be taken into account?

- It is also suggested to underline the novelties that the proposed method offers and to elaborate more on the future work paths.

Comments on the Quality of English Language

 Moderate editing of the English language is required.

Author Response

#Review 3:

The article deals with a very interesting topic regarding the ‘Advanced Farming Strategies Using NASA-POWER Data in Peanut-producing Regions without Surface Meteorological Stations’. Overall, it is a comprehensive article and the findings provided indicate that a great deal of effort was put in. Suggestions for improvements that could be performed to the manuscript prior to its publication are the following:

Abstract:

-It is suggested to avoid acronyms, numerical data and parentheses in this part of the manuscript. Try to keep the Abstract simple yet informative and focus on the research questions that the authors try to answer through the paper, as well as the novelties/results that this paper presents. In addition, the content through lines 30-31 needs to be rephrased.

We change the topic with you said. Some acronyms, parentheses and numerical data were excluded, but in other cases is important insert the values, for example, the name of the variables used in the work. Because the name of variables are very big and we prefer to insert the acronym. In the end of abstract we rephrased the lines.

Introduction:

-It would be interesting to add some corresponding references about the situation in other countries and resume what has been done so far in this field. In addition, it is suggested to add some more comments regarding the novelties that this paper presents. What research gaps does this paper try to fill up regarding previous similar studies?

In this step, I add some information’s about the surface weather stations in another countries around Brazil. Some states and countries have a network of the weather stations for example United States, but in the major of the cases, the countries doesn’t have weather stations. Especially in the peanut production monitor the climate conditions is decisive to improve the production and find the best moment to digging. The peanut plant have a strong correlation with the temperature and degree-days accumulations (obtained using the temperature). Some novelties of this work is evaluate the orbital platform to monitor the climatic conditions and find the best variables that correlation with peanut maturity index (PMI). The NASA-POWER platform can be use by the farmers doesn’t have surface weather stations close the farm in investigate the climatic conditions and make-decision quickly.

- Regarding the citation of your references: Try to use the method proposed by the journal throughout the whole manuscript (keep the same style and make sure you follow a uniform pattern, see lines 84-85 for instance).

Materials and Methods

-Line 83 Typo: 'Figura' -> Figure (please check all captions throughout the manuscript as well and make sure you follow a uniform pattern). Check spelling and English grammar.

We check the grammar and follow a uniform pattern in all text to show the name of the figures and tables.

-Lines 142-149: Any comments about the accuracy of the data collected (given by the providing platform maybe)?

In the line 142 – 149 we prefer discuss how the data were collect and how the platform it work, always compare the platform with the surface weather station. The accuracy of the platform were discuss in the results and discussion, but a new topic were add in the paragraph to improve the characteristics about the NASA-POWER platform. In the NASA-POWER docs (site), it has showed the validation of the meteorology and energy fluxes data. Some variables with temperature have a high RMSE as the values around 3 °C and others as surface pressure the RMSE values are low around 2.87 kPa. This difference was inserted in the paragraph to each studied variable.

-Line 197: It would be useful to your readers to elaborate more on PCA analysis (the same goes for regression analysis -line 244)

In this part of the text was insert a topic with five lines about PCA. PCA can be used to decrease the number of the variables and identify the variables that present a good adjust with the main variable, in this case the PMI. Nevertheless, the PCA can show an idea about the importance of the variable and show how the variables behave in relation with the main variable, for example, the degree-days accumulation follow the PMI, i.e. when the degree-days accumulation increase the PMI increase too.

For the linear regression analysis, the models created in this analysis was used to evaluate the adjustment of the NASA-POWER models in relation the surface weather station. When the values of accuracy (RMSE) and precision (R2) were low and high respectively, the NASA-POWER models had an excellent performance in estimate the variable.

- It is suggested to add a general workflow showing what steps were made, in order to facilitate reading.

This suggested is very important, in the ended of the materials and methods were developed a flowchart the present the main steps of the work. The flowchart was divided in data collect and data analysis and in each step was possible understand how the data were collected and interpreted.

Conclusions

- Lines 459-460: It is suggested to rephrase the sentence.

The sentence was rephrase. The wind speed variable were the worst variable evaluate in this work, but, when this variable was divided by county were possible to understand that the Dougherty showed the worst values and consequently the overall models wasn’t showed a good fit.

- It would be interesting to comment on the following: What could be some possible restrictions or challenges if the proposed methods were implemented in a different area of interest? Are there any other factors that should be taken into account?

The platform can by apply in any areas of interest, in agriculture the tool became an excellent tool in monitor the climate and show information about the climate to the farmers. Not only agriculture but in others areas is possible to use the platform with accuracy and precision. Some restrictions about the platform were found in your accuracy in measure some variables with relative humidity and the extensive size of the grid in collect the data. When the grid is very size, the areas overlap and measure not is difference between the areas.

- It is also suggested to underline the novelties that the proposed method offers and to elaborate more on the future work paths.

The work evaluate the accuracy and precision of the platform in estimate the climatic variables. The countries that did not have access or financial conditions in buy a surface weather stations to monitor the climatic conditions in your country can be use the orbital platform to monitor. The farmers can use the platform to understand the behavior of the crop in different climatic conditions and improve the management and make-decision. In the future with the arrived of the complex mathematical models and artificial intelligence the platforms have the opportunity in grow up more and obtained the data more accuracy and precision.

Round 2

Reviewer 1 Report

Comments and Suggestions for Authors

Accepted for me 

Author Response

#Reviwer1

Comments and Suggestions for Authors: Accepted for me

Authors: thank you for considering our manuscript.

Best regards!

Reviewer 3 Report

Comments and Suggestions for Authors

Introduction

Lines 56-58: Rephrase the sentence.

In addition, you mention Argentina and Mexico but what about other countries? It is still suggested to add relevant references at this point even if there are no available corresponding data. For example, how did the other countries try to surpass this issue?

Check again spelling and English grammar more carefully throughout the whole manuscript. Check again all captions thoroughly.

For example:

line 237 -> Flowchart 1--->Rename to Figure 1

line 256 -> Figura 2 ---> Rename to Figure 2

line 466 -> Figura 9a ----> Rename to Figure 9a

.....etc

Incorporate your comments regarding the last two points (restrictions/application in other areas of interest/ novelties)  more clearly throughout the 'Conclusions'

Comments on the Quality of English Language

Moderate editing of English language is required.

Author Response

#Reviwer2

Comments and Suggestions for Authors:

1) Introduction: Lines 56-58: Rephrase the sentence. In addition, you mention Argentina and Mexico but what about other countries? It is still suggested to add relevant references at this point even if there are no available corresponding data. For example, how did the other countries try to surpass this issue?

In the revised version of the manuscript (lines 56-58), we have modified the sentences to address the issue. The World Meteorological Organization (WMO) recommends the use of 6.3 weather stations per 100 km2; however, the majority of countries do not meet this requirement due to various challenges in establishing and maintaining such stations. Building and installing weather stations are complex tasks that involve technical expertise for data collection and periodic equipment maintenance.

One common approach adopted by countries is the utilization of orbital platforms for monitoring and collecting climatic data. This alternative becomes particularly valuable for nations lacking a sufficient number of ground-based weather stations. Countries with an extensive network of weather stations can leverage orbital platforms to analyze meteorological historical series, calibrate models, create predictive models for future climatic conditions, and calculate essential climatic indices such as evapotranspiration. This integrated approach not only facilitates improved water management but also enhances agricultural practices.

2) Check again spelling and English grammar more carefully throughout the whole manuscript. Check again all captions thoroughly.

For example:

line 237 -> Flowchart 1--->Rename to Figure 1

line 256 -> Figura 2 ---> Rename to Figure 2

line 466 -> Figura 9a ----> Rename to Figure 9a.....etc

All ‘figura’ were rename to ‘Figure’ in the text, included the ‘flowchart’.

Conclusion:

3) Incorporate your comments regarding the last two points (restrictions/application in other areas of interest/ novelties) more clearly throughout the 'Conclusions'

The conclusion of the text has been revised to incorporate details on restrictions, applications, areas of interest, and novelties. The identified limitations in the text pertain to the platform's errors in certain variables, such as relative humidity and wind speed.

The platform proves valuable for monitoring climatic changes, managing agriculture, calculating growing degree days, analyzing evapotranspiration, and determining climatic indices, thereby enhancing agricultural practices.

It is important to note that the platform is accessible free of charge and exhibits a commendable level of accuracy. This accessibility proves particularly beneficial for farmers, especially those involved in peanut cultivation, enabling them to closely track and manage their crops. The platform assists farmers in determining optimal times for planting or harvesting, contributing to more informed decision-making in agricultural activities.